# A two-dimensional mid-infrared optoelectronic retina enabling simultaneous perception and encoding

Fakun Wang[1,6], Fangchen Hu[1,2,6], Mingjin Dai[1], Song Zhu[1], Fangyuan Sun[1], Ruihuan Duan ®[3], Chongwu Wang[1], Jiayue Han ®[1], Wenjie Deng[1], Wenduo Chen[1], Ming Ye[1], Song Han[1], Bo Qiang[1], Yuhao Jin[1], Yunda Chua[1], Nan Chi[2], Shaohua Yu[4], Donguk Nam[1], Sang Hoon Chae ®[1], Zheng Liu ®[3] & Qi Jie Wang ®[1,5] ✉

Infrared machine vision system for object perception and recognition is becoming increasingly important in the Internet of Things era. However, the current system suffers from bulkiness and inefficiency as compared to the human retina with the intelligent and compact neural architecture. Here, we present a retina-inspired mid-infrared (MIR) optoelectronic device based on a two-dimensional (2D) heterostructure for simultaneous data perception and encoding. A single device can perceive the illumination intensity of a MIR stimulus signal, while encoding the intensity into a spike train based on a rate encoding algorithm for subsequent neuromorphic computing with the assistance of an all-optical excitation mechanism, a stochastic near-infrared (NIR) sampling terminal. The device features wide dynamic working range, high encoding precision, and flexible adaption ability to the MIR intensity. Moreover, an inference accuracy more than 96% to MIR MNIST data set encoded by the device is achieved using a trained spiking neural network (SNN).

Infrared (IR) machine vision that can efficiently perceive, convert, and process the massive amount of IR optical information of the observed objects has become an important technology for various scenarios requiring crucial decisions, which include autonomous driving, intelligent night vision, military defense and medical diagnosis[1,2]. The current IR machine vision systems usually rely on physically separated IR imaging devices and von-Neumann computing architectures to perform the real-time information perception and processing, respectively[1,2]. This system generates large amounts of redundant data being exchanged between sensory terminals and processing units, resulting in high data latency, large computing load and low energy efficiency[3–5]. The lack of compactness and computing efficiency is rapidly making the existing system obsolete in the era of big data and the internet of things.

In contrast to the inefficient machine vision system, the human visual sensory system consists of a very compact retina that can perceive, encode and process a huge visual data set by harnessing distributed and parallel neural networks. In the real world, continuous light stimuli are first received by the sensory neurons in the human retina and then encoded as discrete spike trains generated via a set of neural algorithms[6,7]. These encoded spike trains are subsequently transmitted to the visual cortex of the brain for information processing[8,9]. The discretization and stochasticity of spike-encoded information allow long-distance communication and efficient neural computation[8]. Following the infrastructure and operation mechanism of human visual sensory system, it is highly desired to have the perception and encoding of external optical stimuli integrated in one

[1]School of Electrical and Electronic Engineering, Nanyang Technological University, Singapore 639798, Singapore. [2]Key Laboratory for Information Science of Electromagnetic Waves (MoE), Fudan University, Shanghai 200433, China. [3]School of Materials Science and Engineering, Nanyang Technological University, Singapore 639798, Singapore. [4]Peng Cheng Laboratory, Shenzhen 518055, China. [5]Centre for Disruptive Photonic Technologies, School of Physical and Mathematical Sciences, Nanyang Technological University, Singapore 637371, Singapore. [6]These authors contributed equally: Fakun Wang, Fangchen Hu. ✉e-mail: qjwang@ntu.edu.sg

neuromorphic device for realizing a compact, efficient, and intelligent IR machine vision system.

2D van der Waals (vdWs) heterostructures become the promising candidates for achieving such a goal due to their superior optical functionalities such as strong light-matter interaction, tunable bandgap and the potential compatibility with CMOS platform[10,11]. Recently, notable progress has been made with 2D vdWs heterostructures in developing neuromorphic sensors, encoders and processors[12–18], presenting a development trend towards all-in-one devices with functionalities integration[19,20]. However, these studies focus only on the visible and near-infrared (NIR) spectral ranges, while such integrated neuromorphic devices operating in the MIR range would greatly advance IR machine vision systems for autonomous driving, intelligent night visions, defense, and medical applications, and improve the versatility of neuromorphic systems. In addition, the demonstrations of encoding functionality in previous studies are limited in electronic approaches with electrical bias[8,17,19–22]. An integrated MIR neuromorphic device with the perception and encoding functionalities driven by an all-optical approach is expected to shed light on the technological development of high-speed and zero-bias information coding of IR machine vision.

In this work, we report an all-optical driving 2D MIR optoelectronic retina with simultaneous perception and encoding functionalities without inducing electrical bias. The neuromorphic 2D vdWs heterostructure composed of b-AsP and MoTe$_2$ is designed such that it can perceive external light in the MIR spectral range (at ~4.6 µm) while simultaneously encode the received MIR information into spike trains by harnessing a stochastic NIR sampling terminal (at ~730 nm excitation). Featuring high MIR detectivity (9.6 × 10$^8$ cm Hz$^{0.5}$/W) and fast NIR photoresponse rate (~600 ns), the device successfully demonstrates a typical neural encoding algorithm of rate-based encoding with wide dynamic working range and high encoding precision for MIR

illumination intensities. Our device demonstrates the adaption ability to intensity variation of MIR signal, which is analog to the human eye's visual adaption to the change in ambient light intensity in the visible range. Furthermore, a trained SNN achieves an inference accuracy of more than 96% to the MIR MNIST data set which is encoded into spikes by the device. The retina-inspired 2D MIR optoelectronic device integrating perception and encoding functionalities has the potential to perform MIR machine vision in a highly compact and efficient way.

## Results

### Human visual system and the 2D MIR optoelectronic retina

The visual system is one of the important sensory organs for humans to perceive the external world as more than 80% of the environment information is captured in human eyes[10,22]. Figure 1 shows the implementation of perception, encoding and processing of stimulus signals from external objects in the human visual system (top) and presents the proposed 2D MIR optoelectronic device that can mimic the key functionalities (bottom). For the human visual system, the external stimulation signals are perceived by photoreceptors and converted into electrical impulses (spikes) by ganglion cells following neural encoding algorithms, and eventually transmitted to the visual cortex in the brain for processing[8]. Notably, the encoding process exhibits the inherent stochasticity which is involved in the spike generation and enhances the noise tolerance of spikes. Inspired by the human visual system, in this work, a 2D optoelectronic retina capable of simultaneously perceiving and encoding MIR optical stimuli is proposed and demonstrated by using a 2D b-AsP/MoTe$_2$ vdWs heterostructure. Upon the stimulation of MIR signals, the photo-excited current ($I_{DS}$) of the 2D optoelectronic device is measured from source/drain electrodes at zero bias, which mimics the optical signal collection and conversion of the photoreceptors in the human retina. Meanwhile, programmable NIR optical pulses with stochastic intensity cause corresponding

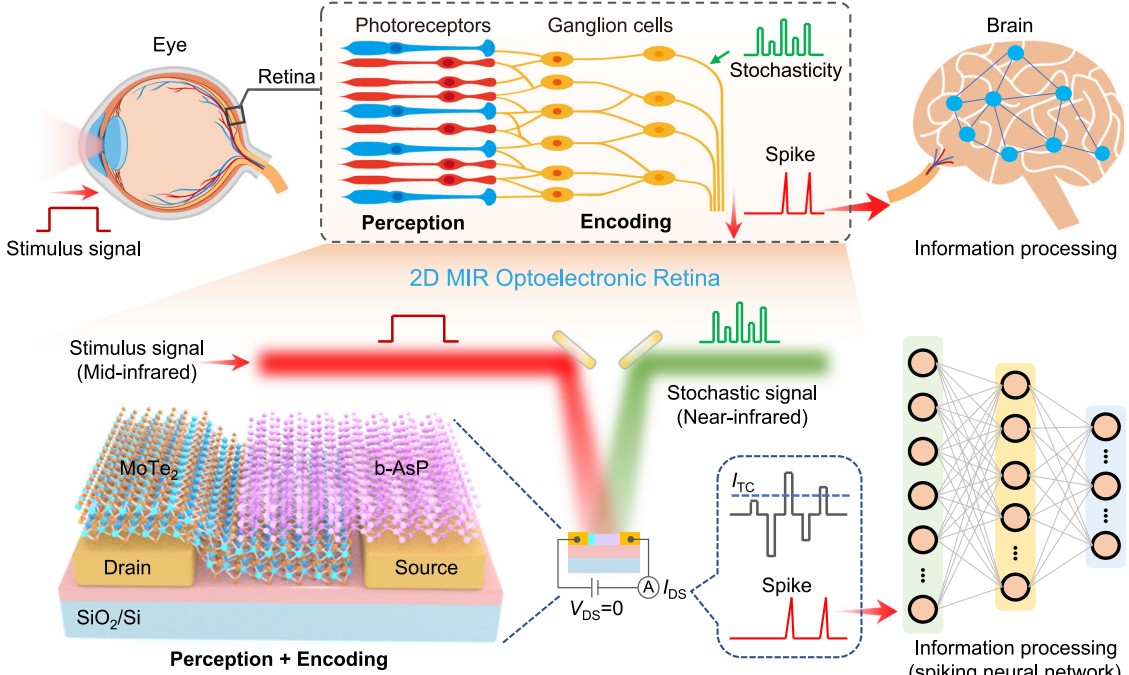

**Fig. 1 | Schematic of human visual system and the proposed 2D MIR optoelectronic retina.** In the human visual system, the stimulus signal from external object entering the eye is first converted into corresponding graded potentials by the photoreceptors (rod and cone cells) in the retina. The potentials are then encoded into spikes by ganglion cells with the participation of inherent stochasticity (random signal) exhibited in sensory transduction. Finally, the spikes with coded stimulus information are transferred to the visual cortex in brain for further

information processing. In the proposed optoelectronic retina (lower), the 2D b-AsP/MoTe$_2$ vdWs heterostructure mimics the retina to realize integrated perception and encoding functionality for MIR objects with the help of a random NIR-light terminal. The encoded spike signal can be obtained from source-drain current ($I_{DS}$) under the threshold current ($I_{TC}$) serving for the input of spiking neural network for information processing.

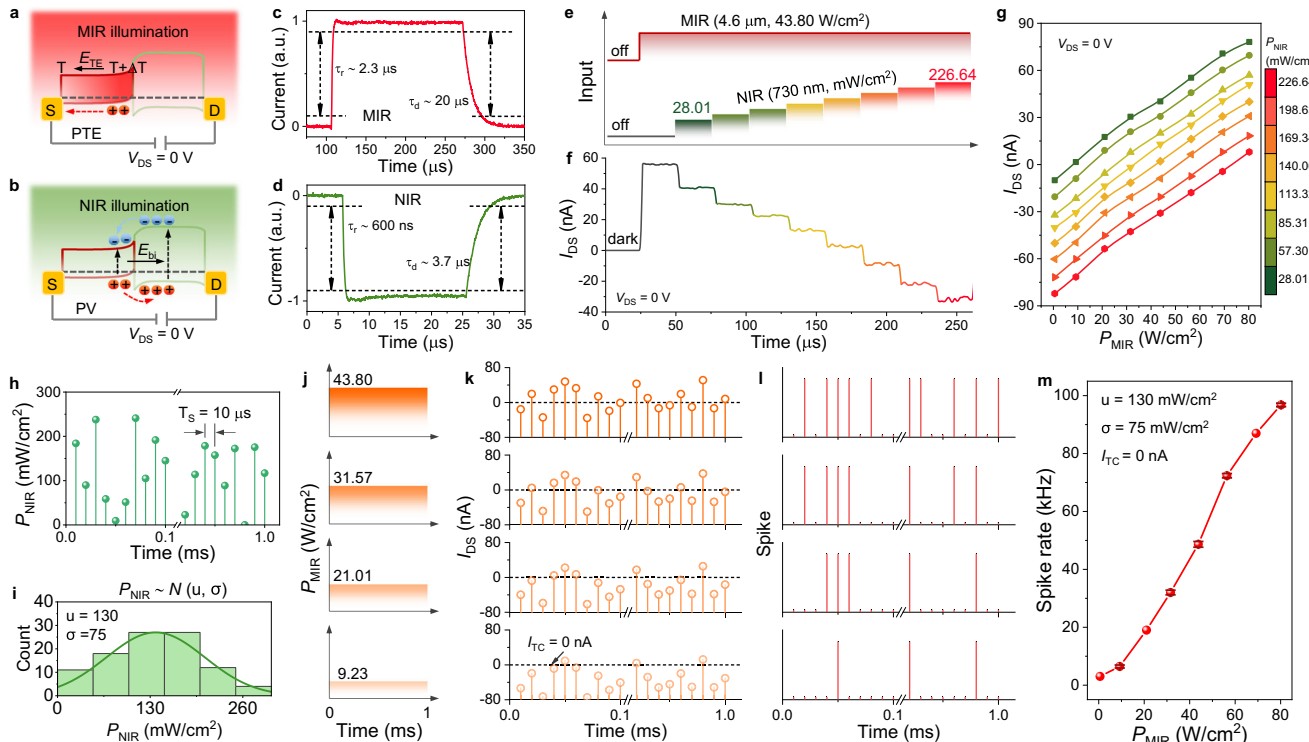

**Fig. 2 | Perception and encoding characteristics of the 2D MIR optoelectronic retina. a, b** Photoresponse mechanisms of the b-AsP/MoTe$_2$ heterostructure under MIR and NIR global illumination at $V_{DS}$ = 0 V. **c, d** Photoresponse rate of the heterostructure at $V_{DS}$ = 0 V. The NIR (730 nm) and MIR (4.6 μm) photoresponse rate of the heterostructure are as fast as 600 ns/3.7 μs and 2.3 μs/20 μs, respectively. **e, f** Output photocurrents ($I_{DS}$) of the heterostructure under an input of simultaneous MIR and NIR illuminations. The power densities of MIR and NIR are shown in (**e**). **g** The dependence of output photocurrent on the MIR power density under the modulation of NIR light with various power densities. **h** One train of NIR optical pulses (100 time-steps (pulses) for one train) that are randomly sampled from a Gaussian distribution for spike rate-based encoding. **i** The Gaussian distribution of NIR power densities with mean of $u$ = 130 mW/cm$^2$ and a standard deviation of $σ$ = 75 mW/cm$^2$. **j** Analog value of MIR power density. **k** Corresponding time-domain transduction current ($I_{DS}$) waveform output from the source electrode for each MIR power density in (**j**). The sampling rate for NIR light is 100 kHz. **l** Corresponding spike train for each $I_{DS}$ waveform in (**g**) when the spike threshold current ($I_{TC}$) is set to 0 nA. The $I_{DS}$ higher than $I_{TC}$ could stimulate one spike. **m** Mean spike rate as a function of $P_{MIR}$ when $u$, $σ$, $I_{TC}$ are 130 mW/cm$^2$, 75 mW/cm$^2$, 0 nA, respectively. Error bars in (**m**) represent the variation (standard deviation) of spike rate.

fluctuation of $I_{DS}$, where a spike is generated when the $I_{DS}$ exceeds the threshold line ($I_{TC}$), emulating the encoding scheme of ganglion cells. The as-generated spike trains with coded MIR information are finally processed by a trained SNN for intelligent tasks, such as classification and decision[8,16,17].

## Perception and encoding characteristics of the 2D MIR optoelectronic retina

In the proposed 2D MIR optoelectronic retina, the b-AsP is used as the MIR photosensitive layer owing to its narrow bandgap of ~0.15 eV and high MIR optical absorption efficiency of ~10%[23,24], and MoTe$_2$ with an appropriate bandgap of ~1.0 eV serves as the NIR sensitizer[25,26]. Both b-AsP and MoTe$_2$ exhibit high hole mobility of ~145 and ~15 cm$^2$/Vs (Supplementary Figs. 1–4), respectively, allowing for fast photoresponse of the b-AsP/MoTe$_2$ devices. The NIR and MIR photoresponse characteristics are discussed in the Supplementary Information (see Supplementary Figs. 1–12 and Note 1 and 2), where the photovoltaic (PV) and photothermoelectric (PTE) effects are identified as the dominant mechanisms for perceiving NIR and MIR illumination, respectively. The schematic diagram of the photocurrent generation in the b-AsP/MoTe$_2$ device under MIR and NIR global illumination are depicted in Fig. 2a, b, respectively. Under MIR laser global illumination, an unbalanced lattice temperature distribution is generated in b-AsP layer due to the asymmetric contacts of b-AsP with MoTe$_2$ and Au electrode. The lattice temperature of b-AsP at the MoTe$_2$ contact side is higher than that at Au electrode contact side because the Seebeck coefficient of b-AsP (723.66 μV/K, see Supplementary Fig. 9) is higher

than that of MoTe$_2$ (142.59 μV/K, see Supplementary Fig. 10) and the thermal conductivity of MoTe$_2$ (~40 W/mK)[27,28] is lower than that of Au (~200 W/mK)[29]. Such lattice temperature distribution promotes the diffusion of holes in the b-AsP from the MoTe$_2$ contact side to Au electrode contact side, thus forming a positive PTE photocurrent under zero bias with b-AsP as the source terminal. Under NIR laser global illumination, both b-AsP and MoTe$_2$ layers generate electron-hole pairs which are separated by the built-in electrical field with direction pointing from b-AsP to MoTe$_2$ side at the junction. The photo-generated electrons and holes move toward b-AsP and MoTe$_2$, respectively, which contributes to the negative photovoltaic photocurrent. As shown in Fig. 2c, d, the NIR and MIR photoresponse rate of the heterostructure are as fast as 600 ns/3.7 μs and 2.3 μs/20 μs, respectively. The asymmetric response time may be caused by the trapping of photo-excited charge carriers by the defect state in the junction interface or by phosphorus oxide on the b-AsP surface[30–32]. Moreover, the detectivity of the device to MIR illumination can reach up to ~9.6 × 10$^8$ cm Hz$^{0.5}$/W. More details and discussions on the photoresponse performance under MIR and NIR illumination are provided in Supplementary Figs. 13–19 and Note 3.

For encoding operations, the MIR stimulus signals and NIR sampling terminal are simultaneously input onto the device. We first demonstrate the photoresponse of the device under simultaneous illuminations of both MIR and NIR laser. As shown in Fig. 2e, f, distinct output photocurrents ($I_{DS}$) can be observed when the device is simultaneously illuminated by MIR with a certain power density and NIR with various power densities. The photoresponse under the

simultaneous illuminations shows high repeatability and stability, evidenced by multiple and reproducible switching (Supplementary Fig. 20). Figure 2g depicts the dependence of $I_{DS}$ on the MIR illumination intensity at different NIR power densities, which is an important reference to obtain dynamic encoding range for MIR power density ($P_{MIR}$) once the $I_{TC}$ and NIR power density ($P_{NIR}$) distribution are given. More important, the stable photoresponse can be still maintained under NIR illumination with a frequency of 100 kHz (Supplementary Fig. 21). Such a fast and stable response makes it possible to generate higher spiking rates and provides a guarantee for high-precision MIR intensity coding.

Next, we experimentally demonstrate the function of simultaneous perception and spike rate-based encoding for $P_{MIR}$. The NIR laser is applied as sampling pulses with amplitude following a Gaussian distribution with a sampling period ($T_S$) of 10 μs (on/off = 5/5 μs), which is analogous to inherent stochasticity[8]. This sampling period is determined by taking into account the NIR response rate. Figure 2h shows a train of NIR optical pulse that is randomly sampled from a Gaussian distribution with the mean, $u = 130$ mW/cm$^2$, and standard deviation, $\sigma = 75$ mW/cm$^2$ for spike rate-based encoding (Fig. 2i). When the NIR sampling pulse and MIR light with a specific intensity are simultaneously illuminated on the device, the response corresponding to each $P_{MIR}$ (Fig. 2j) is recorded by $I_{DS}$. The $P_{MIR}$ is encoded by one train of NIR optical pulses (100 time-steps for one train) and therefore results in a train of $I_{DS}$ with 100 sampling points. As-recorded $I_{DS}$ trains with $I_{TC} = 0$ nA and corresponding spike trains are shown in Fig. 2k, l, respectively. The delineation rule of $I_{TC}$ is discussed in Supplementary Fig. 24. The $I_{DS}$ value higher than $I_{TC} = 0$ nA stimulates one spike. Average spike rate for each $P_{MIR}$ is calculated according to the generated spike train (spike rate $= \frac{1}{T_s} \cdot \frac{n}{\text{Time-steps}}$ (Hz), where $n$ is the number of spikes in the output spike train), as shown in Fig. 2m. It can be clearly observed that the device is capable of simultaneously perceiving and encoding the $P_{MIR}$ within ~80.21 W/cm$^2$. The error in spike rate is about 0.9% due to the fluctuation of $I_{DS}$ waveform. Notably, a fast response speed to NIR light for our device is helpful to increase time-steps over a fixed encoding time which equals the multiplication of time-steps and $T_S$. Insufficient time-steps for one MIR intensity cannot guarantee high encoding accuracy (analyzed in Supplementary Fig. 25).

**Visual adaption ability of the 2D MIR optoelectronic retina**
Adaption occurs in all sensory systems to help them efficiently encode external stimuli as the stimuli distribution changes[33]. For example, the human eyes can identify objects both in starlight and in sunlight by changing neural encoding strategy during the adaption process[34]. For intelligent MIR vision tasks, a high-performance MIR optoelectronic retina should also have such visual adaption ability to satisfy various application scenarios. Two related aspects of the visual adaption ability, namely, dynamic working range and encoding precision are discussed here. A high dynamic working range allows the device to respond to the MIR targets with distinct $P_{MIR}$ difference. For example, the temperature of pig iron and steel strips in industrial process is 427 K and 1457.85 K, respectively[35]. Their $P_{MIR}$ differs over a dynamic range of -24 dB if they are regarded as two ideal blackbodies according to Plank's radiation law[36]. To identify them at the same time, a dynamic working range of $P_{MIR}$ over 24 dB is required. However, the wide dynamic working range sacrifices the encoding precision defined as the resolution of spike rate for unit $P_{MIR}$ in encoded images. The dynamic working range is hence required of compression to attain high encoding precision for some cases that the details of $P_{MIR}$ distribution inside targets need to be accurately identified, such as MIR imaging of human body for medical diagnosis[37].

To demonstrate the adaption ability of our MIR optoelectronic retina, we establish a testing setup shown in Fig. 3a. A metal mask with nine hollow figures "3" illuminated by MIR laser is used to imitate the real MIR targets. The mask can move along the $x$ and $y$ axis to allow MIR light to pass each target in order. By adjusting the output optical power of MIR laser, the $P_{MIR}$ distribution of each target "3" is different. The real $P_{MIR}$ distribution of nine targets "3" is measured by photocurrent mapping method (seen in "Methods" section) and presented in Fig. 3b. For convenience, nine targets "3" are named as (i) to (x) in the incremental order of $P_{MIR}$. To encode the $P_{MIR}$ distribution of targets into corresponding spike trains, another NIR light whose $P_{NIR}$ is sampled from a Gaussian distribution with $u$ and $\sigma$ of 130 and 75 mW/cm$^2$ is also incident into the device at the same time. The recognized image after rate encoding by our device is shown in Fig. 3c. The correlation coefficient (CC) which refers to the similarity of the encoded target and original one, all exceed 97% for targets (i–x) (Bottom curve of Fig. 3f), validating that our device has an excellent encoding precision. This is attributed to the fast response reaching 100 kHz that provides sufficient rate encoding resources for high $P_{MIR}$ resolution.

The adjustment of $u$ and $\sigma$ for sampling the $P_{NIR}$ can conveniently tune the dynamic working range. The increase of $\sigma$ extends the dynamic working range, while the increase of $u$ shifts the dynamic working range to a high $P_{MIR}$ range. The experimental and simulation results are presented in Fig. 3d and Supplementary Fig. 26a–c, respectively. Such dependence can also be observed from the encoded images in Fig. 3e and CCs in Fig. 3f in different cases of ($u$, $\sigma$). For example, when the ($u$, $\sigma$) changes from (70, 35) to (130, 35), the dynamic working range shifts to the high $P_{MIR}$ range, which results in the correct encoding of the high-power target (ix) with CC improving from 83% to 98% but failed encoding of the low-power target (ii) with CC = 0. When the $\sigma$ is increased from 35 to 75 at $u = 130$, the CCs of targets (ii) and (ix) both reach 98% without any encoding failure, which verifies the function of $\sigma$ used to extend dynamic working range. The $u$ should keep a high power when $\sigma$ is relatively high (like $\sigma = 75$ here). Otherwise, the background noise of the encoded image will be magnified due to the no-zero spike rate at $P_{MIR} = 0$, such as the results at ($u$, $\sigma$) = (70, 75), causing an extra interference for identifying targets. To magnify the details of $P_{MIR}$ distribution inside one certain target, a high encoding precision is required and can be achieved by decreasing $\sigma$ under a suitable $u$. For example, the target (ii) at ($u$, $\sigma$) = (70, 35) has a higher contrast than the case at ($u$, $\sigma$) = (70, 75). Therefore, optimizing the $u$ and $\sigma$ values is critical in achieving a suitable dynamic working range and high encoding precision and help exhibit the eye's visual adaption ability to different MIR targets in our device.

**Encoding a perceived image for classification using spiking neural network**
Lastly, we utilize the device to encode the MIR MNIST data set into spike trains, which enables the successful realization of SNN-based digit classification tasks with inference accuracy of more than 96% (see the "Methods" section for the details about preparing the MIR MNIST data set). Compared to traditional artificial neural network (ANN)[38], SNN is believed to be a more efficient neural network that rarely requires high-precise multiplication. Also, the density of binary spikes required for SNN is much sparser than that for ANN, mitigating the storage memory and energy requirements[38]. The energy-delay product of SNN running on a spike-based neuromorphic hardware has been proved by four-orders magnitude lower than that of the traditional DNN running on a CPU over one batch size[39]. We use the snnTorch platform introduced by Eshraghian[38] to establish a fully-connected three-layers SNN that consists of the input layer, hidden layer and output layer with 784, 200 and 10 neurons, respectively, as shown in Fig. 4a. Each image in the MIR MNIST data set with a size of 28 × 28 pixels is perceived and encoded by our device into 784 spike trains that concurrently enter into the input layer of a trained SNN. The training and parameters optimization methods for SNN are described in the Methods section. The 10 spiking neurons in the output layer shown in Fig. 4a represent digits from 0 to 9. The spiking neuron producing the spike train with the highest spike rate corresponds to the digit that

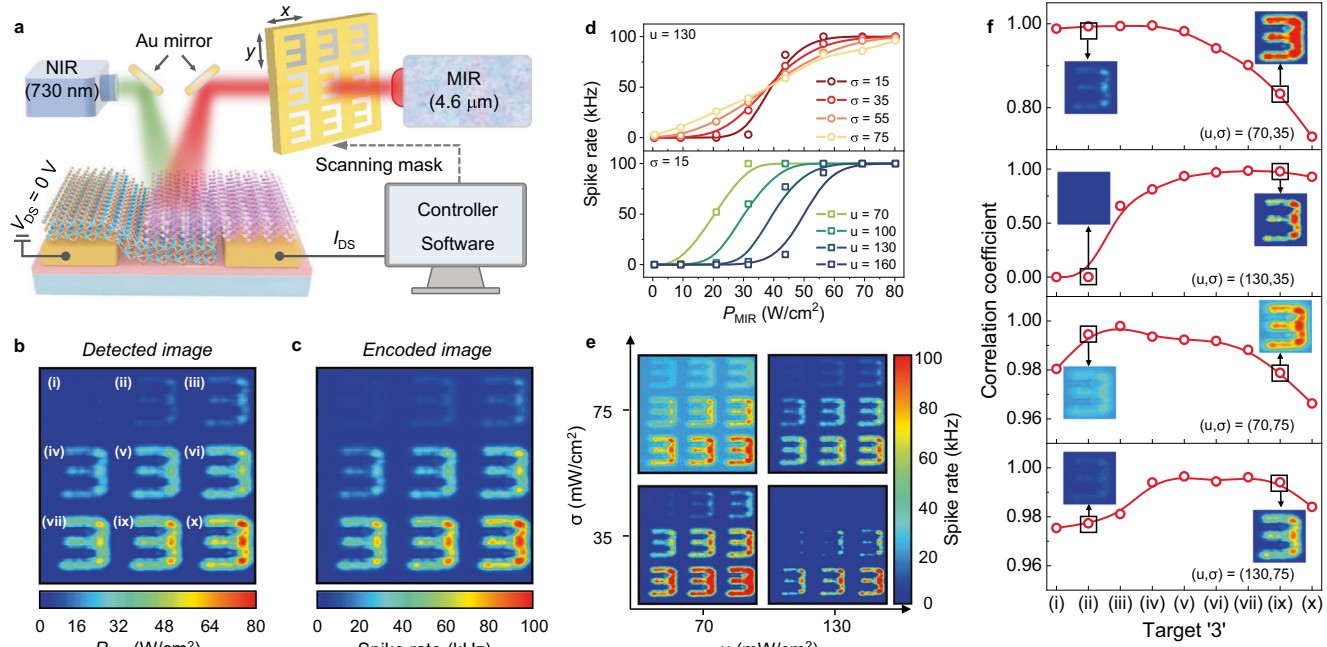

**Fig. 3 | Visual adaption of the 2D MIR optoelectronic retina for MIR targets with different optical power. a** Schematic diagram of test setup for the optoelectronic retina to perceive and encode the MIR targets. A 2D-moveable metal mask provides nine "3"-shaped MIR targets whose average optical power density ($P_{MIR}$) are linearly distributed within 0 to 80.21 W/cm² by adjusting the output optical power of 4.6 μm laser. **b** The directly detected images of nine MIR targets "3", namely (i) to (x) in order. The color distribution is linearly mapped to $P_{MIR}$. **c** The encoded images of nine MIR targets, whose color distribution is linearly mapped to spike rate ranging from 0 to 100 kHz. The rate of every pixel is calculated based on the spike train with 100 time-steps under $u$, $\sigma$, $I_{TC}$ of 130 mW/cm², 75 mW/cm², 0 nA, respectively. **d** Experimental result of spike rate as a function of $P_{MIR}$ for different $u$ and $\sigma$ used for sampling 730 nm light. **e** Results of encoding images under different $u$ and $\sigma$. Such parameter adjustment allows device have flexible dynamic working range and encoding precision to adapt targets with different $P_{MIR}$. **f** The correlation coefficient between each encoded target "3" in (**e**) and their original image in (**b**). A higher correlation coefficient indicates higher encoding precision. Four encoding cases with different ($u$, $\sigma$) are discussed.

SNN predicts. Each spiking neuron in every layer is described by a leaky integrated-and-fire (LIF) neuron model[40], as shown in Fig. 4b. The input pre-neuronal spikes $X_i(t)$ of the $i$th spiking neuron are modulated by synaptic weights $W_i$ to produce a resultant current $\sum_{i=1}^{k} W_i^T X_i(t)$, which affects the membrane potential $V_{mem}$ of the post-neuron in the next neuron layer, given as:

$$V_{mem}(t+1) = \beta V_{mem}(t) + \sum_{i=1}^{k} W_i^T X_i(t) - R\left[\beta V_{mem}(t) + \sum_{i=1}^{k} W_i^T X_i(t)\right] \tag{1}$$

$$R = \begin{cases} 1, & \text{if } V_{mem} > V_{TH} \\ 0, & \text{otherwise} \end{cases} \tag{2}$$

where $\beta$ and $k$ are membrane potential decay rate and the number of neurons in this layer, respectively. The $T$ is the transposition operation. The $V_{mem}$ of the post-neuron will integrates incoming spikes until it reaches membrane threshold $V_{TH}$ where the $V_{mem}$ is reset to zero. Meanwhile, the post-neuron generates an output spike which acts as the input spike of next neuron layer. In our device, $I_{TC}$ is equivalent to the $V_{TH}$.

The classification performance of SNN significantly depends on the dynamic working range and encoding precision of the device. As mentioned in Fig. 3, the $u$ and $\sigma$ values of Gaussian distribution for sampling NIR light control the dynamic working range and encoding precision. If the dynamic working range mismatches the $P_{MIR}$ range of the target within $[0, P_{max}]$ or the encoding precision is insufficient, the inaccurate translation of the target by encoded spikes will cause inference error of SNN. Figure 4c, d shows the

classification accuracy of SNN when the $P_{max}$ of MIR MNIST test set varies from 0 to 80.21 W/cm² at different values of $u$ and $\sigma$. A relatively low $\sigma$ of 35 makes the dynamic working range too narrow to encode the digits with $P_{max}$ lower than 10 W/cm², resulting in 9.8% classification accuracy. When $\sigma$ increases to 55, the enlarged dynamic working range can cover both low and high $P_{max}$ and allows the classification accuracy to become higher than 96%. However, the further increase of $\sigma$ to 75 decreases the encoding precision. The spike rate resolution is not sufficient to support accurate classification for the low-$P_{max}$ case. Additionally, the background noise is a little magnified, hampering the inference of SNN. The $u$ value controls the position of the dynamic working range, and it therefore controls the position of high-accuracy working range of SNN. For example, the working range with classification accuracy higher than 96% gradually moves to higher $P_{MIR}$ range when $u$ increases from 100 to 130 with $\sigma = 15$, shown in Fig. 4d. The time-steps, representing the number of sampling points for NIR light to encode one MIR intensity, also influences the classification accuracy of SNN. As shown in Fig. 4e, classification accuracy increases as the increase of time-steps, and reaches 96% at the time-steps of 100 at an optimal ($u$, $\sigma$) = (70, 25) to encode the target with $P_{max}$ of 21 W/cm². The performance of our device is already comparable to an ideal linear encoder. However, insufficient time-steps result in inadequate representation of targets, and therefore significantly decline the classification accuracy. The encoded images of target "3" using time-steps of 1, 5 and 100 are given in Fig. 4e (i–iii), respectively, which highlights the significance of sufficient time-steps for accurate encoding and inference of SNN. The results of the accuracy vs. time-steps for other $P_{max}$ are also provided in Supplementary Fig. 28, which suggests low-power MIR objects require more time-

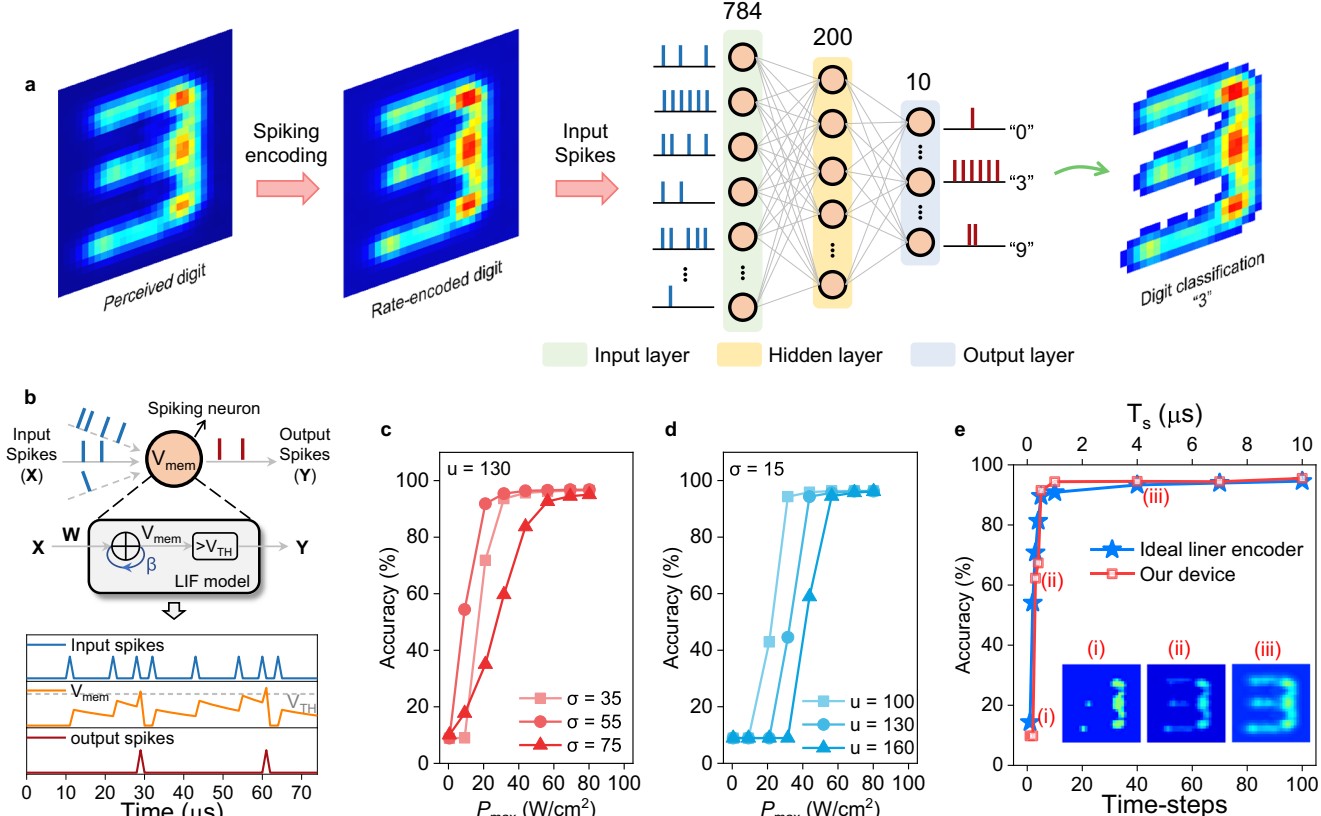

**Fig. 4 | Digit encoding and classification by the 2D MIR optoelectronic retina and SNN. a** The MIR digit image is detected and rate-based encoded into several series of spike trains which enter a trained fully-connected SNN to realize digit classification task. The corresponding digit of the output neuron having the highest spike rate is the predicted result. **b** The leaky integrate-and-fire (LIF) neuron model used in each nodes of SNN. The membrane voltage ($V_{mem}$) increases with the weighted input spikes ($\boldsymbol{W} \cdot \boldsymbol{X}$) until it reaches a constant threshold ($V_{TH}$) at which an output spike appears in the output spike train ($\boldsymbol{Y}$) and the $V_{mem}$ is reset to zero. During the period without input spike, the $V_{mem}$ decays with the membrane

potential decay rate ($\beta$) of 0.95. Classification accuracy of SNN vs. different $P_{max}$ of MIR MNIST images at different $\sigma$ and $u$ is described in (**c**) and (**d**), respectively. The $u$, $I_{TC}$ and time-steps in (**c**) are 130 mW/cm$^2$, 0 nA and 100, respectively. The $\sigma$, $I_{TC}$ and time-steps in (**d**) are 15 mW/cm$^2$, 0 nA and 100, respectively. **e** Classification accuracy of SNN vs. time-steps of one spike train when $P_{max}$ is 21 W/cm$^2$. The $\sigma$ and $u$ are optimized to 25 and 70 mW/cm$^2$, respectively, with the $I_{TC}$ of 0 nA. Insets: (i–iii) show the encoded images of "3" when time-steps are 1, 5 and 100, respectively. Too few sampling points result in an inadequate representation of digits after encoding.

steps to achieve accurate classification compared to high-power MIR objects. These facts indicate a fast response speed to NIR light in our device is critical to help SNN realize accurate MIR objects classification using short encoding time. Besides, the impact of device thicknesses, different wavelengths and distribution of the sampled stochastic light on encoding precision and classification accuracy of SNN are also discussed in Supplementary Figs. 29–31. Overall, by optimizing the encoding parameters, our device can ensure the fast and accurate encoding ability on MIR objects, as well as help SNN realize MIR objects classification tasks with the inference accuracy up to 96%.

## Discussion

Inspired by the human vision system with the function of perceiving, transmitting and processing the external environment information, we demonstrate a compact retina-inspired MIR optoelectronic device using a 2D b-AsP/MoTe$_2$ vdWs heterostructure. The device features a high MIR (~4.6 μm) detectivity of 9.6 × 10$^8$ cm Hz$^{0.5}$/W and a fast NIR (730 nm) response rate of ~600 ns without inducing electrical bias. Impressively, the proposed device could not only perceive the MIR illumination stimuli, but also encode it into rate-based spike trains with the assistance of a stochastic NIR sampling terminal. Moreover, device's encoding range and precision can be flexibly adjusted for different MIR illumination intensities. The device encodes the MIR MNIST data set into spike trains which enables SNN

to achieve digit classification with an accuracy higher than 96%. Our work provides a promising routine for constructing compact and efficient MIR neuromorphic devices for night machine vision, military, defense, and medical diagnosis. We anticipate that the optical approaches of realizing neuromorphic functions based on 2D vdWs heterostructures have the potential of wide bandwidth up to tens of gigahertz when combined with integrated guided-wave nanophotonics[25,41], bringing in the advantages of low data latency and high energy efficiency.

## Methods

### Device fabrication and characterization

Because 2D b-AsP and MoTe$_2$ flakes are sensitive to the water and oxygen in the surrounding environment, a dry transfer method was applied to fabricate the 2D b-AsP/MoTe$_2$ vdWs heterostructure. The contact electrodes (5/50 nm Cr/Au) were first patterned on a SiO$_2$/Si substrate by standard photolithography and electron beam evaporation. The exfoliated 2D b-AsP and MoTe$_2$ flakes from bulk crystals were then dry transferred onto the electrodes. Finally, h-BN encapsulation was used to protect the device from degradation. The morphology and thickness of as-fabricated device were characterized by optical microscope (Nikon), atomic force microscope (Bruker Dimension Icon). Scanning photocurrent mapping was performed by using confocal micro-Raman spectroscopy (WITec alpha300) equipped with a focused 532 nm laser.

**Article**

## Detection and encoding measurements

The measurements of electrical and photoelectric properties were performed at room temperature and under ambient air conditions. A digital source meter (Keysight, B2912A) was used to apply voltage to the device and record the generated current. A MIR quantum cascade laser (QCL) (Daylight Solution, MIRCat) with tunable wavelength from 3.5 to 11.0 µm was employed as the external stimuli. The power of MIR laser was recorded by a thermal power meter (OPHIR, Nova display-ROHS). A power adjustable 730 nm laser (HÜBNER Photonics, Cobolt 06-MLD) was applied as the stochastic terminal and its power density was measured using a power meter (Thorlabs, PM100D). The laser spots of MIR laser and 730 nm laser are about 100 µm, which is larger than the size scaling of the as-fabricated 2D b-AsP/MoTe$_2$ vdWs heterostructure. For the encoding measurements, the device is simultaneously illuminated by 4.6 µm MIR laser with a fixed power density and pulsed 730 nm laser with Gaussian distribution power densities. The sampling period ($T_S$) of 730 nm laser is set to 10 µs and its amplitude is determined by the desired encoding algorithm. The fast current sampling was collected by means of an oscilloscope (Keysight, DSOX3054T).

## Photocurrent mapping method to recognize $P_{MIR}$ distribution image

To recognize the $P_{MIR}$ distribution image of figure "3" targets in mask, the responding photocurrent of device to every pixel of mask is collected by oscilloscope. The mask has $300 \times 300$ pixels in which each "3" target occupies $100 \times 100$ pixels. The $P_{MIR}$ of 4.6 µm laser from QCL on every "3" region ($100 \times 100$ pixels) is different. When the mask is scanned by pixels, the responding photocurrent of each pixel depends on the optical flux of 4.6 µm laser passing through this pixel region. According to the mapping relation of photocurrent and $P_{MIR}$ given in Supplementary Fig. 16b, the corresponding $P_{MIR}$ for every pixel can be estimated from the photocurrent obtained by experiment, and finally constitutes the $P_{MIR}$ distribution image shown in Fig. 3b.

## Preparation of MIR MNIST data set

The MIR MNIST data set is obtained by mapping pixel values of traditional MNIST data set ranging in [0, 255] to optical power density of 4.6 µm laser ranging in [0, $P_{max}$]. Once the $P_{max}$ is set, every image in the prepared MIR MINST data set with a size of $28 \times 28$ pixels is first flatten to obtain 784 analog optical power density of MIR laser. The MIR laser with a certain optical power density can be detected and encoded by our device into spike trains as the input of SNN.

## Training and parameters optimization of SNN

For training of SNN, a surrogate gradient descent algorithm is used to update synaptic weights[38] in order to avoid dead neuron problem. The loss function and optimizer used here are cross-entropy loss and Adam optimizer. There are 60,000 and 10,000 MIR MNIST images used for training and test, respectively. The number of hidden neurons and membrane potential decay rate are two super-parameters affecting classification ability of SNN. More hidden neurons and higher $\beta$ can enhance the classification accuracy (seen in Supplementary Fig. 27a, b). The $\beta$ of real synaptic devices hardly reaches 100%, and therefore the $\beta$ in our work is set to 0.95. The number of hidden neurons is set to 200 considering the trade-off between performance and complexity. After training around 450 iterations in one epoch with the batch size of 128, the loss of train and test sets all converge to a steady level, verifying SNN is well trained without under-fitting and over-fitting problems (seen in Supplementary Fig. 27c).

## Data availability

The data that support the findings of this study are available within the main text and Supplementary Information. Any other relevant data are available from the corresponding author upon reasonable request. Source data are provided with this paper.

## Code availability

The code can be available from the corresponding author upon reasonable request.

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

## Acknowledgements
This work was supported by the Singapore Ministry of Education (MOE-T2EP50120-0009 (Q.J.W.)), Agency for Science, Technology and Research (A*STAR) (A18A7b0058 (Q.J.W.) and A2090b0144 (Q.J.W.)), National Medical Research Council (NMRC) (MOH-000927 (Q.J.W.)), and National Research Foundation Singapore (NRF-CRP22-2019-0007 (Q.J.W.)), National Key Research and Development Program of China (2022YFB2802803 (N.C.)), the Natural Science Foundation of China Project (61925104 (N.C.), 62031011 (N.C.)) and Major Key Project of PCL (N.C.), and F.H. acknowledges the support from the China Scholarship Council.

## Author contributions
F.W. and F.H. designed the experiments and analyzed the data. F.W. and F.H. wrote the manuscript. F.W., F.H. and M.D. fabricated the devices. S.Z. performed the atomic force microscope measurements. F.S., R.D., C.W., J.H., W.D., W.C., M.Y., S.H., B.Q., Y.J., and Y.C. provided experimental testing support. D.N., S.H.C., Q.J.W., N.C., S.Y. and Z.L. revised the manuscript. Q.J.W. supervised the project. All authors have discussed the results and commented on the manuscript.

## Competing interests
The authors declare no competing interests.
