## [Peer Review File · Nature Communications]

REVIEWER COMMENTS

Reviewer #1 (Remarks to the Author):

I went through the response letter and I feel that the authors have addressed my concerns. Now I support its publication in Nature Communications.

Reviewer #2 (Remarks to the Author):

In this revised manuscript, the authors have fully addressed the questions from reviewers. I would recommend acceptance in the present form.

Reviewer #3 (Remarks to the Author):

Although the authors have addressed some of my concerns in the revised manuscript. However, some key questions are not yet to be addressed. The specific questions are provided below.

1. The authors showed that the variation of thickness has little impact on the performance of the proposed device, which is within the expectation. However, my question is that how the reduction in lateral size of the heterostructure would affect the coding and recognition accuracy. If the mechanism is right, then the performance may be drastically reduced since the noise plays more important role in the device with reduced size. Besides, the verification of the sub-500 nm length in channel would be critical to potential in high-density integration.

2. Although the authors demonstrated that increasing the light intensity would enhance the recognition accuracy, the light intensity is on the order of W/cm^2 and is two orders of magnitude

larger than the nominal value of solar constant 0.137 W/cm^2 on the earth. Therefore, the proposed device or technology is impractical to application since the recognition accuracy is very poor at such light level.

3. The device exhibits asymmetric rising and falling time, why is that? Such asymmetric time scale would impose a limitation to the sampling rate.

4. I was still unconvinced by the authors about the importance of the MIR retinomorph device, since the human retina only responds to the visible light. Besides, the proposed technology is more complex than the mature technology based on the thermal imaging and DNN, since the practical implementation of the proposed device relies on the peripheral circuit to set the threshold value for encoding and requires additional NIR source.

5. Based on the proposed device, it seems that the device itself does not play a unique role in generating the current spikes, because the generation of spikes relies on the external NIR light pulses. This indicates that many other previously reported MIR devices can be used for this demonstration.

6. Since the device does not share similar features, therefore the words such retina, perception, appearing in the title, should be removed to avoid misleading the authors of interest.

Reviewer #1 (Remarks to the Author):

I went through the response letter and I feel that the authors have addressed my concerns.
Now I support its publication in Nature Communications.

Response: Thanks for the reviewer's affirmation to the paper. We are grateful to the reviewer for his/her support.

Reviewer #2 (Remarks to the Author):

In this revised manuscript, the authors have fully addressed the questions from reviewers. I would recommend acceptance in the present form.

Response: Many thanks to reviewer for appreciating the paper. We really appreciate the reviewer's recommendation of acceptance in the present form.

Reviewer #3 (Remarks to the Author):

Although the authors have addressed some of my concerns in the revised manuscript. However, some key questions are not yet to be addressed. The specific questions are provided below.

Response: We appreciate the reviewer for providing the constructive comments. The point-to-point responses to the concerns are provided as seen in the following replies.

1. The authors showed that the variation of thickness has little impact on the performance of the proposed device, which is within the expectation. However, my question is that how the reduction in lateral size of the heterostructure would affect the coding and recognition accuracy. If the mechanism is right, then the performance may be drastically reduced since the noise plays more important role in the device with reduced size. Besides, the verification of the sub-500 nm length in channel would be critical to potential in high-density integration.

Response: We thank the reviewer for this comment. We agree with the reviewer that the reduction of lateral size would reduce the photoresponse performance and recognition accuracy due to the decreased light absorption efficiency and prominent noise. As for the verification of sub-500 nm channel length, we believe it is not necessary because such a channel length is shorter than the diffraction limit of mid-infrared wavelength, which will make it difficult to achieve mid-infrared detection. For instance, the pixel size of commercial infrared cameras is around ten microns, such as 12 μm for vanadium oxide [1], 15 μm for HgCdTe [2], and 20 μm for InGaAs [3], which indicates that our device with the lateral size of $\sim 10 \mu\text{m}$ is consistent and compatible with the commercial mid-infrared cameras.

2. Although the authors demonstrated that increasing the light intensity would enhance the recognition accuracy, the light intensity is on the order of W/cm^2 and is two orders of magnitude larger than the nominal value of solar constant $0.137 \text{ W}/\text{cm}^2$ on the earth.

Therefore, the proposed device or technology is impractical to application since the recognition accuracy is very poor at such light level.

Response: We thank the reviewer for this comment. The major contribution of our work is the first presentation and demonstration of a retina-inspired optoelectronic device that can achieve simultaneous perception and encoding functionalities in the MIR region using an optical excitation. We didn't expect the first demonstration will achieve a device performance which can be commercially exploited immediately in practical applications in challenging environments. On the other hand, we have seen great potential of the proposed mechanism in which the recognition accuracy performance for low MIR light intensity can be significantly improved through the possible approaches listed below, after the work is published and more efforts from the community are invested. Furthermore, many scenes in real life have higher radiation intensity, such as fire site, smelter and so on, where our current devices can already be readily adopted and utilized in such relatively high temperature environments that are difficult for the human body to access.

The recognition accuracy of the targets with low light intensity can be improved from two aspects. One is to optimize the mean (μ) and standard deviation (σ) of the sampling pulses by using a NIR laser with high output resolution, high power stability, and low noise. Another is to design new device structure for higher photoresponse performance. For example, combining meta-surfaces with 2D materials is an effective method to improve the performance due to the enhanced near-field intensity and light absorption induced by the plasmon resonance [4-6]. This improvement will require additionally substantial efforts for structural simulation and device fabrication, which goes beyond our original intention of demonstrating simultaneous MIR perception and encoding in a single device. We will make efforts in this direction in our future work.

3. The device exhibits asymmetric rising and falling time, why is that? Such asymmetric time scale would impose a limitation to the sampling rate.

Response: We thank the reviewer for carefully reading our manuscript. This asymmetric response time may be caused by the trapping of photo-excited charge carriers by the defect state in the junction interface or by phosphorus oxide on the b-AsP surface, resulting in a longer falling time than the rising time [7-9]. As noted by the reviewer, this asymmetric timing does limit the sampling rate. In our work, the longer falling time has been taken into account for coding demonstrations.

We added the following statements to the revised manuscript: “The asymmetric response time may be caused by the trapping of photo-excited charge carriers by the defect state in the junction interface or by phosphorus oxide on the b-AsP surface”. “This sampling period is determined by taking into account the NIR response rate.”

4. I was still unconvinced by the authors about the importance of the MIR retinomorph device, since the human retina only responds to the visible light. Besides, the proposed technology is more complex than the mature technology based on the thermal imaging and DNN, since the practical implementation of the proposed device relies on the peripheral circuit to set the threshold value for encoding and requires additional NIR source.

Response: We thank the reviewer for this comment. Retinomorph optoelectronic device that can perform information perception and pre-processing near or within the sensor is an important innovation platform for intelligent machine vision in the era of big data and the internet of things. Although considerable efforts have been made to constructive retinomorph optoelectronic devices operated in the visible range and have demonstrated their capabilities of information pre-processing, such devices cannot be used in the environments at night or when sunlight is unavailable. As we know, information is not only from the visible range, but also other frequency bands including the infrared. We therefore believe that the retinomorph optoelectronic device is not necessarily restricted to solely processing the visible information perceived by human vision. Expanding the processing range to the infrared spectrum can be of great benefit for advanced intelligence machine vision. Mid-infrared (MIR) regime, an important

frequency band for night vision, sensing, spectroscopy, and free-space communications, carries environment information that can't been visually perceivable by humans. To this end, developing MIR retinomorphic device is of great significance to intelligent machine vision.

In our proposed scheme, a single device that can simultaneously perceive and encode MIR illumination intensities integrates the function of sensor and encoder, which reduces the complexity of machine vision imaging system to a certain extent and caters to the current pursuit of neuromorphic devices. As for the peripheral circuit and NIR light source needed for future practical applications of this device, current mature integrated circuit technology and commercial Vertical-Cavity Surface-Emitting Laser (VCSEL) technology can provide effective solutions, respectively. The threshold setting can be achieved by just adding signal amplification and comparison functions to the readout circuit necessary for any electronic device, which is not a challenge for the current mature microelectronics integration technology. In addition, the VCSEL technology has already been widely used in smartphones (e.g. iPhones) for facial recognition with the advantages of easy integration, low cost, mass production, compatible packaging with integrated chips and CMOS technologies. The emitting diameter of individual VCSEL can be shrank to be $\sim 10 \mu\text{m}$, which is comparable to the lateral size of our device. Moreover, the van der Waals materials b-AsP and MoTe_2 possess potential compatibility with any substrates due to their dangling-bond-free advantage. Therefore, a scheme composed of VCSEL and b-AsP/ MoTe_2 van der Waals device not only reduces the integration complexity, but also takes full advantages of their individual characteristics, which would provide a feasible technology routine for future applications.

5. Based on the proposed device, it seems that the device itself does not plays unique role in generating the current spikes, because the generation of spikes relies on the external NIR light pulses. This indicates that many other previously reported MIR device can be used for this demonstration.

Response: Thanks for the comments. Indeed, the generation of spikes depends on the NIR light pulses. However, the proposed device is linked with the NIR light for perceiving and encoding the external MIR signals. That is, the device plays the role of converting the received MIR signal into a NIR-modulated signal, which is a prerequisite for the generation of current spikes. Moreover, the device itself takes low energy consumption due to the PTE and PV photoresponse mechanisms under zero bias. Therefore, excellent MIR photoresponse that can be modulated by fast NIR light pulses and zero-bias operation mechanism are the unique features of our device in achieving simultaneous perception and encoding functions. As far as we know, the aggregation of these features in a single MIR device has not been reported in previous reports. In addition, most of the reported MIR devices can only be used for perceiving illumination intensity, but can't simultaneously encode the intensity due to the lack of optically tunable property.

6. Since the device does not share similar features, therefore the words such retina, perception, appearing in the title, should be removed to avoid misleading the authors of interest.

Response: We thank the reviewer for this comment. In the human retina, two main functions i.e. perception and encoding are implemented by photoreceptors and ganglion cells [10-12], respectively. In our work, these main functions have been successfully realized by using a dedicatedly designed device. Moreover, our device features the adaption ability to intensity variation of MIR signal, which is analogue to the human eye's visual adaption to the change in ambient light intensity. We therefore believe that the demonstrated device presents similar features to the retina without inducing other misleading information. On the other hand, some other most recent works that mimic the retina's functions also used "retina" and "perception" words. For example "A flexible capacitive photoreceptor for the biomimetic retina. *Light Sci. Appl.* 11, 3 (2022)", "All-in-one two-dimensional retinomorph hardware device for motion detection and recognition. *Nat. Nanotechnol.* 17, 27-32, (2021)", "Bioinspired in-

sensor visual adaptation for accurate perception. *Nat. Electron.* **5**, 84-91, (2022)".

References:

- [1] <https://www.gst-ir.net/products/uncooled-infrared-detectors/>
- [2] <https://www.teledyneimaging.com/en/aerospace-and-defense/products/sensors-overview/infrared-hgcdncte-mct/>
- [3] <https://www.hamamatsu.com/us/en/product/cameras/ingaas-cameras.html>
- [4] Ren, Z. *et al.* Wavelength-multiplexed hook nanoantennas for machine learning enabled mid-infrared spectroscopy. *Nat. Commun.* **13**, 3859, (2022).
- [5] Venuthurumilli, P. K. *et al.* Plasmonic resonance enhanced polarization-sensitive photodetection by black phosphorus in near infrared. *ACS Nano* **12**, 4861-4867, (2018).
- [6] Li, W. *et al.* Harvesting the loss: surface plasmon-based hot electron photodetection. *Nanophotonics* **6**, 177-191, (2017).
- [7] Bullock, J. *et al.* Polarization-resolved black phosphorus/molybdenum disulfide mid-wave infrared photodiodes with high detectivity at room temperature. *Nat. Photon.* **12**, 601-607, (2018).
- [8] Ahmed, T. *et al.* Fully light-controlled memory and neuromorphic computation in layered black phosphorus. *Adv. Mater.* **33**, 2004207, (2021).
- [9] Liu, C. *et al.* Polarization-resolved broadband MoS₂/black phosphorus/MoS₂ optoelectronic memory with ultralong retention time and ultrahigh switching ratio. *Adv. Funct. Mater.* **31**, 2100781, (2021).
- [10] Subbulakshmi Radhakrishnan, S. *et al.* A biomimetic neural encoder for spiking neural network. *Nat. Commun.* **12**, 2143, (2021).
- [11] Wu, Q. *et al.* Spike encoding with optic sensory neurons enable a pulse coupled neural network for ultraviolet image segmentation. *Nano Lett.* **20**, 8015-8023, (2020).
- [12] Chen, W. *et al.* Retinomorphie optoelectronic devices for intelligent machine vision. *iScience* **25**, 103729, (2022).

REVIEWERS' COMMENTS

Reviewer #3 (Remarks to the Author):

I have no other questions.

REVIEWERS' COMMENTS

Reviewer #3 (Remarks to the Author):

I have no other questions.

Response: We thank the reviewer for his/her constructive comments and suggestions on this work.